# Effect of Soil Salinity and Foliar Application of Jasmonic Acid on Mineral Balance of Carrot Plants Tolerant and Sensitive to Salt Stress

Sylwester Smoleń *[ID], Aneta Lukasiewicz[ID], Magdalena Klimek-Chodacka[ID] and Rafal Baranski[ID]

Department of Plant Biology and Biotechnology, Faculty of Biotechnology and Horticulture, University of Agriculture in Krakow, AL. 29 Listopada 54, 31-425 Krakow, Poland; aneta.lukasiewicz1@gmail.com (A.L.); magdalena.klimek-chodacka@urk.edu.pl (M.K.-C.); rafal.baranski@urk.edu.pl (R.B.)
* Correspondence: sylwester.smolen@urk.edu.pl

**Abstract:** The aim of the study is to determine the effects of soil salinity stress and foliar application of jasmonic acid (JA) on the mineral balance in plants of salt-sensitive doubled haploid carrot line (DH1) and salt-tolerant local DLBA variety (DLBA). Concentrations of 28 elements were determined in roots and leaves and in the soil. The *DcNHX4* gene (cation:proton exchange antiporter) expression was assessed. The salinity stress reduced the mass of roots and leaves more in DH1 than in DLBA. DLBA plants accumulated larger amounts of Na and Cl in the roots and had an increased transport of these elements to the leaves. The salt-tolerant and salt-sensitive carrot varieties differed in their ability to uptake and accumulate some elements, such as K, Mg, Zn, S, Cd, P and B, and this response was organ-specific. A selective uptake of K in the presence of high Na concentration was evident in the tolerant variety, and a high Na content in its leaves correlated with the expression of *DcNHX4* gene, which was expressed in DLBA leaves only. JA application did not affect the growth of DLBA or DH1 plants. In the sensitive DH1 variety grown under salinity stress, JA induced changes in the mineral balance by limiting the uptake of the sum of all elements, especially Na and Cl, and by limiting Zn and Cd accumulation.

**Keywords:** antiporter; *Daucus carota*; jasmonic acid; macroelements; microelements; NaCl; salt stress; trace elements

## 1. Introduction

It is estimated that up to 20% of 230 million hectares of cultivated, irrigated land is affected by uncontrolled concentration of salt in the soil. The situation is likely to worsen, due to the anticipated effect of global climate warming and advances in agricultural production mainly due to the expansion of areas where irrigation is applied [1]. Excess salt accumulates in the soil at high concentrations and leads to plant growth inhibition mostly by reducing their ability to take up water and by injuries to the cells in transpiring leaves Therefore, salt stress causes similar effects to drought stress with respect to cellular and metabolic processes. In consequence, deleterious effects lead to reductions in yield [2]. In the presence of 200 mM NaCl in the soil, the estimated yield reduction in salt-tolerant (sugar beet) and moderately tolerant (cotton) crops is 20% and 60%, respectively, while salt-sensitive (soybean) plants may die [3]. Thus, soil salinization is a growing problem for agriculture worldwide and poses a serious threat to crop yields and future food production.

The threshold value above which deleterious effects occur can vary depending on several factors including plant type, soil-water regime and climatic conditions [4]. Plants that have developed functional mechanisms of salt tolerance can grow in soils with higher salt concentrations. Plant

tolerance to salinity is defined as the ability to maintain growth in saline conditions relative to the growth in non-saline conditions. Increased soil salt concentrations decrease the ability of a plant to take up water. $Na^+$ and $Cl^-$, taken up in large amounts, negatively affect growth by impairing metabolic processes and decreasing photosynthetic efficiency [5]. To protect against salinity-induced oxidative stress, manifested in the production of several cytotoxic reactive oxygen species (ROS) with hydrogen peroxide as an example, plant cells have developed a complex antioxidant defense system [6]. A key requirement for plant growth under a high concentration of salt is to maintain a high $K^+$:$Na^+$ ratio in the cytosol. It can be realized by diminishing the entry of $Na^+$ ions into the cells, extrusion of $Na^+$ ions out of the cell and $Na^+$ vacuolar compartmentation [7]. The later process is mediated by cation-proton exchanger transporters that may have different specificity to monovalent cations and which are involved in, cytosolic and vacuolar, ion and pH homeostasis, osmotic adjustment, and their activity may influence plant development such as flower development [8–10]. Integral membrane proteins functioning as $K^+$ and $Na^+$ symporters or uniporters play a major role in salt tolerance through regulation of $Na^+$ transport downward from leaves to roots, preventing $Na^+$ over-accumulation in the leaves under salinity stress [5].

Plant phytohormones such as jasmonates and salicylic acid are involved in the adaptation and response of plants to abiotic and biotic stress factors [11]. Jasmonic acid (JA) and its derivative methyl jasmonate (MeJA) are derivatives of the fatty acid metabolism and are synthesized in plants in the octadecanoid pathway [12]. Apart from participating in the response of plants to stress, JA takes an active part in plant growth, storage organ formation, fruit ripening, plant senescence and reproduction, regulation of flowering time, flower morphogenesis and gender differentiation.

JA and MeJA are signaling molecules that induce the activation of various stress-related genes responsible for, inter alia, the functioning of metabolic pathways under stress conditions [13,14]. For example, exogenous application of JA and MeJA was shown to have a positive effect on plants exposed to salinity stress. Foliar application of JA to wheat seedlings enhanced salt stress tolerance by a marked decrease in the concentrations of malondialdehyde and $H_2O_2$ and activation of the superoxide dismutase (SOD) and peroxidase (POD) enzymes and increases in glutathione (GSH) and Chl *b* contents [15]. Pre-treatment with MeJA mitigated the negative effect of salt stress on broccoli [16] and soybean [17].

Genetic resources of cultivated carrot can be separated into two genetically distinct groups, the Eastern (Asian) and Western (European and American) gene pools differing in plant morphology and chemical composition [18,19]. Western carrots appear as a more advanced group, better adapted for commercial production and processing. Eastern carrots, commonly grown in Asia, produce rather thicker, shorter roots, often have pubescent leaves, tend to flower early and are poorer in β-carotene than Western carrots, but may be better adapted to warmer climate [20].

The carrot is classified as one of the most salt-sensitive vegetable species [21]. Other experiments showed that yield of carrot was reduced by about 50% at salinity levels ranging from 70 mM to 100 mM NaCl [22] and by 15% and 37% when plants were irrigated with 40 mM and 80 mM NaCl solutions, respectively [23]. Although cultivated carrot is salt-sensitive its tolerance to high NaCl concentration can be increased by enhancing osmoprotective mechanisms. The introduction of betaine aldehyde dehydrogenase gene into a salt-sensitive carrot variety caused a more than 50-fold increase in the concentration of betaine and, in consequence, such engineered plants tolerated up to 400 mM NaCl in soil [24]. Despite harmful effects of saline conditions on carrot, there are several regions in Asia where landraces adapted to high soil salinity are being cultivated [25]. Plants of those populations are capable of withstanding high salt concentrations due to unknown genetic determinants controlling salt tolerance mechanisms. Those plants can be used to reveal the nature of salt tolerance in carrot and probably in other root crops, at least in the family Apiaceae. They may also serve as a source of new genes for crop improvement by breeding for enhanced salt tolerance in future.

Most studies on salinity stress that focus on plant nutrition processes document the influence of this factor on plant growth and development and on the Na and Cl uptake by plants. Accumulation

of these elements can, however, significantly affect the uptake and concentration of other elements important for the functioning of plants—this issue has not yet been sufficiently examined in carrot.

The aim of our study was to determine the effect of soil salinity stress on the functioning of mineral metabolism in salt-sensitive and salt-tolerant carrot plants. In addition, the study aimed at examining whether and to what extent, the application of exogenous JA affected the mineral nutrition process in carrot plants under salinity stress conditions.

## 2. Material and Methods

### 2.1. Plant Material and Cultivation

Two carrot (*Daucus carota* L. ssp. *sativus* Hoffm.) populations were used, a salt-sensitive doubled haploid line DH1 derived from a western-type Nantes carrot [26] and the Iranian local variety DLBA (Univ. Agric. Krakow collection) grown in the Fars region where saline soil occurs. The latter variety was considered as salt tolerant. The experiment was conducted in a plastic tunnel ($30 \times 12 \times 5$ m length/width/height) with tilted side vent in the roof and located at the University of Agriculture in Krakow campus ($50° 05' 03''$ N $19° 57' 00''$ E; 216 m altitude). The roof was automatically opened when temperature inside the tunnel reached 18 °C/12 °C (day/night) to prevent overheating the plants. Seeds were sown in mid-March in plastic containers ($60 \times 40 \times 41$ cm length/width/height) filled up to 35 cm in height with 84 L of soil mixture (thereinafter called "soil") composed of 1:1 (*v/v*) sand and peat substrate (Biovita, Tenczynek, Poland). The properties of the soil were: pH 6.40, EC 0.2 dS·m$^{-1}$; Eh—redox 280.7 mV and macroelement contents (mg·dm$^{-3}$): 10.6 N-NO$_3$, 25.0 N-NH$_4$, 21.3 P, 90.2 K, 54.2 S, 1260.1 Ca, 89.7 Mg, 15.0 Na and 1.8 Cl. Salt stress was induced by using the soil with EC, adjusted to 3 dS·m$^{-1}$ by mixing with 1.67 g NaCl per liter of soil prior to filling the containers to 25 cm in height, which was then covered with a 10 cm of the soil with EC 0.2 dS·m$^{-1}$ (i.e., the same soil without NaCl addition). Seedlings—and then plants—growing in these containers were irrigated with tap water (EC 0.6 dS·m$^{-1}$) for eight weeks and then the plants growing in the saline soil were watered every 2–3 days with a 100 mM NaCl solution (EC 10.7 dS·m$^{-1}$). In total, the plants were watered with the NaCl solution 22 times, using on average 0.6 L of the solution per container, per day. The watering with NaCl was used to increase the degree of salinity of the top layer of the soil in the containers and to maintain a uniform, constant level of salinity in the soil. The control plants were grown in containers containing the soil with EC 0.2 dS·m$^{-1}$ and were irrigated with tap water throughout the experiment, ensuring the volume of water used was the same as the volume of the NaCl solution applied to the salt-stressed plants. Before the plant top dressing treatment with the NaCl solution, jasmonic acid (JA; Duchefa, Haarlem, The Netherlands) was applied to plants twice, two days apart, by spraying their leaves with a 2 mM JA solution. Each time, the plants were sprayed with 10 mL of JA solution per container.

In order to ensure an optimal plant nutrition status with respect to all macro- and micronutrients, all plants were fertilized three times by top dressing through fertigation. A solution of two fertilizers was used: 0.5% Superba™ Zielona Forte NUTRIFOL™ (18–11–35 + micro) + 0.5% lime saltpeter YaraLiva Calcinit (15.2% N /14.5% N-NO$_3$ + 0.7% N-NH$_4$/ and 27.5% Ca)—both fertilizers produced by Yara and distributed by Yara Poland. Each time, 1.5 L of the solution of both fertilizers was applied per container. The fertigation started in the 11$^{th}$ week of plant growth. The next two fertigation treatments were carried out at 7-day intervals. The experiment was set up in random design with five replications, i.e., five containers per each combination of variety, NaCl treatment and JA application ($2 \times 2 \times 2$), with 13 plants per container. Harvesting of the control plants and those subjected to salinity stress was done at the same time, which was 102 days after sowing. The final EC measured in the soil from the control and NaCl treatments after harvest was 0.22 dS·m$^{-1}$ and 3.15 dS·m$^{-1}$, respectively. All plants in a container were harvested; the leaves and storage roots were weighted separately. Then mean masses of storage root and leaves per plant were calculated as well as mean plant biomass, and these means from five replications were subjected to statistical analyses.

### 2.2. Analysis of Elements in Plants

All fresh leaves and, separately, storage roots of plants from each container were cut and mixed. Samples of approx. 200 g from each replication were dried at 70 °C in a laboratory dryer with forced air circulation and ground in a FRITSCH Pulverisette 14 variable speed rotor mill (Idar–Oberstein, Germany) using 0.5 mm sieve. Samples were subsequently analyzed with respect to the concentrations of the following elements: N (by the Kjeldahl method with the use of a VELP Scientifica UDK 193 distillation unit; Usmate, Italy) [27] and Na, P, K, Mg, Ca, S, B, Cu, Fe, Mn, Mo, Zn, Al, Ba, Cd, Co, Eu, La, Li, Lu, Sm, Sr, Th, Ti, Y and Yb by the ICP-OES technique [28] (using an ICP-OES Prodigy Spectrometer, Leeman Labs, New Hampshire, MA, USA) after sample digestion in 65% super pure $HNO_3$ in a CEM MARS-5 Xpress (CEM World Headquarters, Matthews, NC, USA) microwave system [28]. In order to analyze the Cl− content, samples were subjected to extraction with 2% acetic acid. The chloride content in the samples was estimated by the nephelometric method after reaction with $AgNO_3$ [29]. Chemical analyzes of leaves and storage roots were made in two laboratory replicates, separately for each of the five biologic replicates.

### 2.3. Analysis of Elements in Soil

Soil samples were taken for chemical analyses before carrot seed sowing and after the harvest from each container. The soil samples were extracted for 30 min with 0.03 M acetic acid (1:10; soil: extraction solution ratio *v/v*) [29]. Then in these extracts, concentrations of Na, by the ICP-OES technique and Cl, by the nephelometric method after reaction with $AgNO_3$ [29], were determined. The extracts were also used to determine the concentrations of N /N-$NH_4$ and N-$NO_3$/ by the FIA technique [30,31], as well as P, K, Mg, Ca and S (by the ICP-OES technique). After extraction with a 1 M HCl solution (Rinkis method—[32], the concentrations of B, Cu, Fe, Mn, Mo, Zn, Al, Ba, Cd, Co, Eu, La, Li, Lu, Sm, Sr, Th, Ti, Y and Yb were determined using the ICP-OES technique. Before cultivation, the control and saline soil contained: 15.0 and 1703.9 mg Na·kg$^{-1}$ of soil, as well as 1.8 and 3134.2 mg Cl·kg$^{-1}$ of soil, respectively. The concentrations of all the other elements prior to cultivation were at a similar level in the control and saline soil.

### 2.4. Gene Expression

Leaf and root samples frozen in liquid nitrogen were first ground in a mortar. Then, total RNA was isolated using the Direct-zol$^{TM}$ RNA MiniPrep Plus kit with TRI Reagent (Zymo Research, Irvine, CA, USA), followed by DNA elimination using Turbo DNA-free™ Kit (Invitrogen, Thermo Fisher Scientific, Waltham, MA, USA), according to manufacturer protocols. cDNA was synthesized using 1 µg of RNA and the iScript cDNA Synthesis Kit (Bio-Rad, Hercules, CA, USA).

Polymerase chain reaction (PCR) was performed in 10 µL volumes containing 5 µL of PCR Mix Plus Green (A&A Biotechnology, Gdansk, Poland), 0.5 µL forward and 0.5 µL reverse 10 µM primers and 1 µL of cDNA for the *actin-7* (XM_017386919.1) carrot gene used as a reference or 5 µL of cDNA for the predicted *Dcnhx4* (XM_017396153) gene. The Eppendorf Master Gradient thermocycler program was set up as follow: 94 °C for 2 min, 35 cycles of 94 °C for 30 s, 53 °C (for *actin-7*) or 48 °C (for *nhx4*) for 30 s and 72 °C for 1 min. Amplified products were separated in 1.2% agarose gel and visualized in UV light.

Real time quantitative PCR (qPCR) was performed using the Maxima SYBR Green/ROX qPCR Master Mix (Thermo Fisher Scientific) in three technical replications. Reactions were set up in volumes of 15 µL containing 7.5 µL SYBR, 0.6 µL forward and 0.6 µL reverse 5 µM primers, and 3 µL template cDNA. Thermal cycling conditions of the QuantStudio$^{TM}$ 3 System (Applied Biosystems, Foster City, CA, USA) were as follow: 95 °C for 10 min, 40 cycles of 95 °C for 10 s and 60 °C for 45 s and single product amplification was validated in a melt curve analysis. Expression data were normalized against the *actin-7* gene. Reactions were done in at least six biologic replications. Relative gene expression was calculated using the REST 2009 (Qiagen, Hilden, Germany) software [33].



The following primer pairs were used for the *nhx4* gene:    (5′-3′) NHX-4-318-F CGGGGGGTTATTATGTCTCAC and NHX-4-318-R TTACTGTCTGGTGTCTGACT with the PCR amplified expected fragment length of 318 bp, NHX4-F CTCAAGTGTTCGTGAAGTTG and NHX4-R TTCAGTGACATTGTGCCAT with the expected qPCR fragment length of 145 bp; for the *actin-7* gene: Actin7-F GGTATTGTGTTGGACTCTGGTGAT and Actin7-R AGCAAGGTCAAGACGGAGTATG with the expected fragment length of 95 bp.

*2.5. Statistical Analysis*

The concentrations of all the elements analyzed in storage roots and leaves as well as in the soil are presented in terms of the molar mass of the element and expressed in mmol·kg$^{-1}$ dry matter (DW) of the relevant part of the carrot plant or soil. The use of the same unit made it possible to compare the results in terms of uptake efficiency (in the 'soil-plant' system) and accumulation of individual components in the plants, taking into account their molar mass.

The results obtained were statistically verified using the ANOVA module of Statistica 12.0 PL software. Means were compared using Tukey's test for multiple comparisons at the 0.05 significance level. Means are provided with their standard errors. Frequencies of bolting plants were compared using the test for significance level between two proportions.

**3. Results**

*3.1. Plant Growth*

Most seeds of both DH1 and DLBA populations germinated within 10 days and there were no visible differences between plants growing in the control and saline soil for the first eight weeks after sowing. Then the watering of the plants with the NaCl solution started. With the continued NaCl treatment the leaves of DH1 plants tended to show more severe symptoms of chlorosis and wilting and withered faster than the leaves of DLBA plants. Plants of both varieties differed in root and leaf growth (all significance levels for fresh mass and dry weight of roots, leaves and the whole plant were $p < 0.001$) which was restricted by NaCl treatment (all $p < 0.001$) (Table 1).

**Table 1.** Mean mass (g) of the storage root and leaves and biomass of a single carrot plant.

| JA Application | Variety | Treatment | Storage Root | Leaves | Total Biomass | Total Dry Weight |
|---|---|---|---|---|---|---|
| No JA | – | – | 36.3 ± 4.11a | 21.6 ± 2.21a | 57.9 ± 6.05a | 9.0 ± 0.91a |
| JA | – | – | 35.9 ± 4.46a | 20.1 ± 2.01a | 56.0 ± 6.26a | 8.4 ± 0.92a |
| – | DLBA | – | 41.1 ± 4.25b | 26.8 ± 1.83b | 68.0 ± 5.88b | 10.7 ± 0.86b |
| – | DH1 | – | 31.1 ± 3.95a | 15.0 ± 1.34a | 46.1 ± 5.28a | 6.7 ± 0.73a |
| – | – | Control | 54.2 ± 1.62b | 27.5 ± 1.74b | 81.7 ± 3.11b | 12.1 ± 0.60b |
| – | – | NaCl | 19.8 ± 1.11a | 15.0 ± 1.37a | 34.8 ± 2.40a | 5.6 ± 0.41a |
| No JA | DLBA | Control | 57.8 ± 1.33d | 35.9 ± 1.32c | 93.7 ± 1.92d | 14.4 ± 0.40d |
| No JA | DLBA | NaCl | 23.8 ± 1.33b | 20.1 ± 1.32b | 43.9 ± 1.92b | 7.3 ± 0.40b |
| No JA | DH1 | Control | 48.8 ± 1.33c | 20.8 ± 1.32b | 69.6 ± 1.92c | 10.3 ± 0.40c |
| No JA | DH1 | NaCl | 14.9 ± 1.33a | 9.6 ± 1.32a | 24.5 ± 1.92a | 3.9 ± 0.40a |
| JA | DLBA | Control | 62.1 ± 1.48d | 31.6 ± 1.47c | 93.7 ± 2.15d | 14.3 ± 0.45d |
| JA | DLBA | NaCl | 25.1 ± 1.33b | 20.6 ± 1.32b | 45.7 ± 1.92b | 7.3 ± 0.40b |
| JA | DH1 | Control | 48.6 ± 1.48c | 21.1 ± 1.47b | 69.7 ± 1.92c | 9.5 ± 0.45c |
| JA | DH1 | NaCl | 15.6 ± 1.33a | 9.7 ± 1.32a | 25.3 ± 2.15a | 3.8 ± 0.40a |

Means followed by the same letters for main effects or 3-way interaction are not significantly different. at $p = 0.05$; means ± standard error; $n = 5$; JA—jasmonic acid.

The DH1 plants grown in the non-saline soil developed roots and leaves with, on average, 3.2-fold and 2.2-fold greater mass, respectively, than the salt-stressed plants (Table 1). A more adverse effect of salinity on roots than leaves was also observed for DLBA plants. The DLBA control plants had 2.4-fold and 1.7-fold greater root and leaf mass, respectively, than the stressed plants. Thus, the whole DLBA plant biomass was halved in the saline soil, while the DH1 plant biomass was reduced almost to a third. Analogous response was observed with regard to dry weight reduction. However, independent on the variety, NaCl treatment caused an increase of dry matter content in roots from 11.9% to 13.7%, i.e., by a factor of 1.15, while in leaves dry matter content decreased from 20.4% to 19.2% (×0.94). The less restricted growth of DLBA plants compared with DH1 plants indicated that the cultivar DLBA was more tolerant to salt stress. Some plants of this variety also showed annuality, i.e., they bolted by developing shoots with inflorescences. More bolters were observed for the control than for the salt-stressed plants, 25.6% and 15.3%, respectively ($p = 0.029$). Such plants were excluded from further chemical analyses as they had poorly developed storage roots. None of the DH1 plants bolted. Foliar applications of JA did not affect the growth of DLBA or DH1 plants, either in the control or under stressed conditions (the $p$ value ranged between 0.128–0.625 depending on the organ). Also the frequency of bolters in the DLBA variety remained unaffected. Two-way and three-way interactions between factors were statistically insignificant ($p > 0.2$).

### 3.2. Uptake and Accumulation of Na and Cl in Plants

The DLBA plants accumulated, on average, more Na than DH1 plants (509 and 422 in roots and 745 and 612 mmol·kg$^{-1}$ D.W in leaves, respectively) and more Cl (analogously in roots: 484 and 356; in leaves: 1026 and 745 mmol·kg$^{-1}$ D.W) (all significance levels for this main effect: $p < 0.03$; Figure 1). NaCl treatment significantly increased Na contents in roots from 93 to 801 and in leaves from 131 to 1171 mmol·kg$^{-1}$ D.W. and increased Cl contents in roots from 32 to 769 and in leaves from 307 to 1406 mmol·kg$^{-1}$ D.W (all $p < 0.001$). However, a significant variety x NaCl treatment interaction was found for the content of both elements in both organs ($p < 0.002$). In the control conditions, the DLBA plants had similar abilities to accumulate Na and Cl as DH1 plants, but in the saline soil they accumulated more of both elements. The tolerant variety DLBA grown under salt stress conditions accumulated in the roots 10.3 times more Na and 28.6 times more Cl while in the leaves 12.9 and 5.7 times more of Na and Cl, respectively, than the control (Figure 1). The susceptible DH1 plants exposed to salinity stress accumulated 6.5 times more Na and 3.5 times more Cl in the roots, and 7.3 times more Na and 19.9 times more Cl in the leaves, compared with the control plants. There were significant effects of variety, NaCl treatment and their interaction (all $p < 0.001$) as well as variety x JA and 3-way interactions for Na and Cl uptake by both organs (all $p < 0.02$; except 3-way interaction for Cl uptake in leaves: $p = 0.152$). The uptake of these elements by the whole plant growing under the control conditions was the same in both varieties. However, under the conditions of salinity stress, the Na uptake by DLBA plants was 1.6 times higher than in DH1 plants, and the Cl uptake was 1.9 times higher (Table 2).

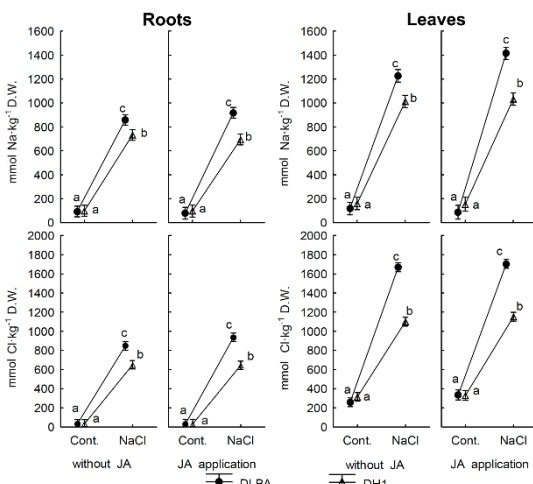

**Figure 1.** Concentrations of Na and Cl in carrot storage roots and leaves. Means followed by the same letters are not significantly different at *p* = 0.05. Whiskers indicate standard error (*n* = 5).

**Table 2.** Uptake of Na and Cl by storage roots, leaves and the whole single plant (mmol per single plant) and the balance of these elements in the "soil-plant" system.

| Element | JA Application | Variety | Treatment | Uptake (mmol) by Single Plant | | | |
|---|---|---|---|---|---|---|---|
| | | | | Storage Root | Leaves | Whole Plant | Balance in Soil–Plant [†] (%) |
| Na | No JA | – | – | 1.73 ± 0.278a | 2.33 ± 0.526a | 4.12 ± 0.209a | 8.27 ± 0.791b |
| Na | JA | – | – | 1.50 ± 0.287a | 2.39 ± 0.416a | 3.57 ± 0.222a | 5.89 ± 0.839a |
| Na | – | DLBA | – | 1.90 ± 0.306b | 3.06 ± 0.563a | 4.77 ± 0.226b | 5.63 ± 0.816a |
| Na | – | DH1 | – | 1.35 ± 0.223a | 1.66 ± 0.264a | 2.93 ± 0.226a | 8.52 ± 0.816b |
| Na | – | – | Control | 0.62 ± 0.082a | 0.70 ± 0.065a | 1.30 ± 0.226a | 13.07 ± 0.839b |
| Na | – | – | NaCl | 2.53 ± 0.196b | 3.86 ± 0.378b | 6.39 ± 0.209b | 1.09 ± 0.791a |
| Na | No JA | DLBA | Control | 0.74 ± 0.196a | 0.81 ± 0.273a | 1.55 ± 0.355a | 12.31 ± 1.390bc |
| Na | No JA | DLBA | NaCl | 2.90 ± 0.196c | 4.79 ± 0.273c | 7.69 ± 0.355d | 1.28 ± 0.390a |
| Na | No JA | DH1 | Control | 0.61 ± 0.196a | 0.70 ± 0.273a | 1.31 ± 0.355a | 18.14 ± 1.390c |
| Na | No JA | DH1 | NaCl | 2.69 ± 0.196c | 3.25 ± 0.273c | 5.94 ± 0.355c | 1.35 ± 0.390a |
| Na | JA | DLBA | Control | 0.56 ± 0.219a | 0.61 ± 0.305a | 1.17 ± 0.397a | 7.86 ± 1.554b |
| Na | JA | DLBA | NaCl | 3.12 ± 0.196c | 5.54 ± 0.273c | 8.66 ± 0.355d | 1.08 ± 0.390a |
| Na | JA | DH1 | Control | 0.53 ± 0.219a | 0.63 ± 0.305a | 1.16 ± 0.397a | 13.96 ± 1.554c |
| Na | JA | DH1 | NaCl | 1.42 ± 0.196b | 1.87 ± 0.273b | 3.29 ± 0.355b | 0.65 ± 0.390a |
| Cl | No JA | – | – | 1.42 ± 0.290a | 3.21 ± 0.481a | 4.73 ± 0.169a | 40.00 ± 2.667a |
| Cl | JA | – | – | 1.34 ± 0.317a | 3.31 ± 0.528a | 4.29 ± 0.174a | 39.66 ± 2.828a |
| Cl | – | DLBA | – | 1.70 ± 0.343b | 4.39 ± 0.557b | 5.91 ± 0.169b | 36.03 ± 2.749a |
| Cl | – | DH1 | – | 1.06 ± 0.233a | 2.13 ± 0.232a | 3.11 ± 0169a | 43.63 ± 2.749a |
| Cl | – | – | Control | 0.21 ± 0.021a | 1.68 ± 0.132a | 1.90 ± 0.174a | 78.59 ± 2.828b |
| Cl | – | – | NaCl | 2.44 ± 0.204b | 4.68 ± 0.467b | 7.12 ± 0.164b | 1.07 ± 2.667a |
| Cl | No JA | DLBA | Control | 0.24 ± 0.169a | 1.84 ± 0.233a | 2.08 ± 0.274a | 68.49 ± 4.737b |
| Cl | No JA | DLBA | NaCl | 2.86 ± 0.169c | 6.49 ± 0.233c | 9.35 ± 0.274d | 1.39 ± 0.737a |
| Cl | No JA | DH1 | Control | 0.21 ± 0.169a | 1.36 ± 0.233a | 1.57 ± 0.274a | 88.92 ± 4.737c |
| Cl | No JA | DH1 | NaCl | 2.38 ± 0.169c | 3.56 ± 0.233b | 5.94 ± 0.274c | 1.19 ± 0.737a |
| Cl | JA | DLBA | Control | 0.22 ± 0.189a | 2.22 ± 0.260ab | 2.44 ± 0.306a | 73.15 ± 5.296b |
| Cl | JA | DLBA | NaCl | 3.18 ± 0.169c | 6.59 ± 0.233c | 9.77 ± 0.274d | 1.08 ± 0.437a |
| Cl | JA | DH1 | Control | 0.17 ± 0.189a | 1.35 ± 0.260a | 1.52 ± 0.306a | 83.80 ± 5.296c |
| Cl | JA | DH1 | NaCl | 1.32 ± 0.169b | 2.08 ± 0.233a | 3.40 ± 0.274b | 0.60 ± 0.237a |

† Percentage ratio of element uptake by all plants from one container to the total concentration of elements. in the soil. Means followed by the same letters for main effects or 3-way interaction are not significantly different at *p* = 0.05; means ± standard error; *n* = 5; JA—jasmonic acid.

The JA-treated DH1 plants exposed to salt stress had a restricted uptake of Na and Cl (by the root, leaves and the whole single plant) in comparison with the plants with no JA application by an average of 45% and 43%, respectively (Table 2). Such response was not observed in the DH1 plants

growing in the control soil. The uptake of Na and Cl by the DLBA plants, including the roots and leaves evaluated separately, growing under the control and salt stress conditions remained unaffected by the foliar JA application.

The increased concentration of NaCl in the soil was the reason that for both varieties the balance of Na and Cl uptake in the 'soil-plant' system was significantly lower than when the plants were growing under the control conditions (Table 2) and the variety x treatment interactions for Na and Cl were also significant ($p < 0.05$). Also the main effect of JA for Na was significant ($p = 0.048$).

### 3.3. Influence of Salinity on the Uptake and Accumulation of Other Elements in Plants

There were significant effects of variety, NaCl treatment and interactions between both factors on the accumulation of all elements analyzed in roots and leaves as well as the uptake of all the analyzed elements by the whole single plant (all $p < 0.05$, except for variety effect on accumulation in roots: $p = 0.277$; Table 3). Plants of both varieties responded to salt stress by higher accumulation of all elements, the sum of the concentrations of all the analyzed elements increased by: 131.3% and 97.8% for DLBA and by 89.0% and 54.2% for DH1, for roots and leaves, respectively (Table 3). This response was independent of JA application. However, the DLBA and DH1 plants had a different mineral metabolism, which depended on their response to soil salinity. In the roots and leaves of DH1, the salt stress caused an increase in P content (51% and 80%, respectively) to a greater extent than in the tolerant DLBA (39% and 59%, respectively) (Figure 2). In addition, the salt stress caused an approx. 20% reduction in the K content of DH1 leaves, which was not evident in DLBA. In the roots, the reduction in the K content was at a similar level in both varieties (by 12% and 16% for DH1 and DLBA, respectively). In DH1 roots, a significant increase (by 36%) in the Mg content was detected, whereas in DLBA roots a slight tendency towards lowering the concentration of this element was observed. In both varieties, salinity had no effect on the Mg content of the leaves.

**Table 3.** Total concentration (mmol·kg$^{-1}$ D.W.) of all the analyzed elements, including Na and Cl, in the storage roots and leaves, as well as uptake of all the analyzed elements by the whole single plant.

| JA Application | Variety | Treatment | Content | | Uptake |
|---|---|---|---|---|---|
| | | | Storage Root | Leaves | Whole Plant |
| No JA | – | – | 2 218.6 ± 54.5a | 4 146.6 ± 272.9a | 26.4 ± 0.76b |
| JA | – | – | 2 191.0 ± 57.8a | 4 305.7 ± 297.4a | 23.0 ± 0.81a |
| – | DLBA | – | 2 160.8 ± 56.1a | 4 350.3 ± 342.1b | 29.2 ± 0.78b |
| – | DH1 | – | 2 248.8 ± 56.1a | 4 102.0 ± 212.1a | 20.2 ± 0.78a |
| – | – | Control | 1 430.5 ± 57.8a | 3 074.5 ± 60.5a | 23.8 ± 0.81a |
| – | – | NaCl | 2 979.1 ± 54.5b | 5 377.7 ± 108.9b | 25.6 ± 0.76a |
| No JA | DLBA | Control | 1 321.5 ± 46.2a | 2 900.1 ± 128.5a | 27.6 ± 2.2cd |
| No JA | DLBA | NaCl | 2 954.2 ± 115.8b | 5 662.3 ± 159.3c | 30.7 ± 1.2d |
| No JA | DH1 | Control | 1 594.8 ± 65.8a | 3 133.4 ± 83.4a | 21.9 ± 1.6bc |
| No JA | DH1 | NaCl | 3 003.9 ± 69.7b | 4 890.6 ± 91.7b | 25.5 ± 1.8bcd |
| JA | DLBA | Control | 1 288.5 ± 126.1a | 2 943.1 ± 108.6a | 26.4 ± 1.7bcd |
| JA | DLBA | NaCl | 3 079.2 ± 213.5b | 5 895.8 ± 92.2c | 32.0 ± 1.8d |
| JA | DH1 | Control | 1 517.5 ± 79.5a | 3 321.5 ± 47.4a | 19.4 ± 0.5ab |
| JA | DH1 | NaCl | 2 879.2 ± 73.8b | 5 062.4 ± 108.1b | 14.2 ± 0.4a |

Total concentration of all the analyzed elements: Na, Cl, N, P, K, Mg, Ca, S, B, Cu, Fe, Mn, Mo, Zn, Al, Ba, Cd, Co, Eu, La, Li, Lu, Sm, Sr, Th, Ti, Y and Yb. Means followed by the same letters for main effects or 3-way interaction are not significantly different at $p = 0.05$; means ± standard error; $n = 5$; JA—jasmonic acid.

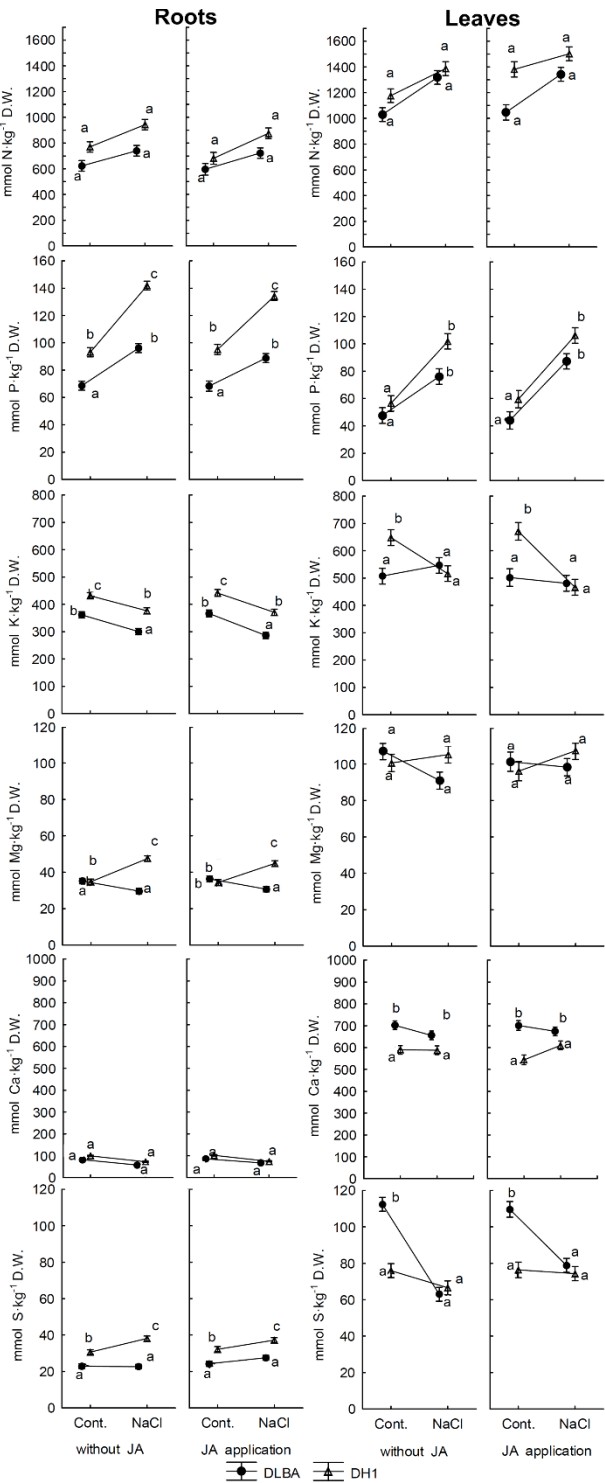

**Figure 2.** Concentrations of N, P, K, Mg, Ca and S in carrot storage roots and leaves. Means followed by the same letters are not significantly different *p* = 0.05. Whiskers indicate standard error (*n* = 5).

DH1 also responded by an increase in B content in the leaves (by 20%) and Zn in the roots (by 36%) of the plants not treated with JA (Figure 3). This variety responded with a much higher accumulation of Cd in comparison with DLBA (Figure 3). On average, the NaCl-treated DH1 plants accumulated 200% more Cd in the roots and 115% in leaves than the control plants while DLBA plants accumulated 92% and 109% more Cd in the roots and leaves, respectively.

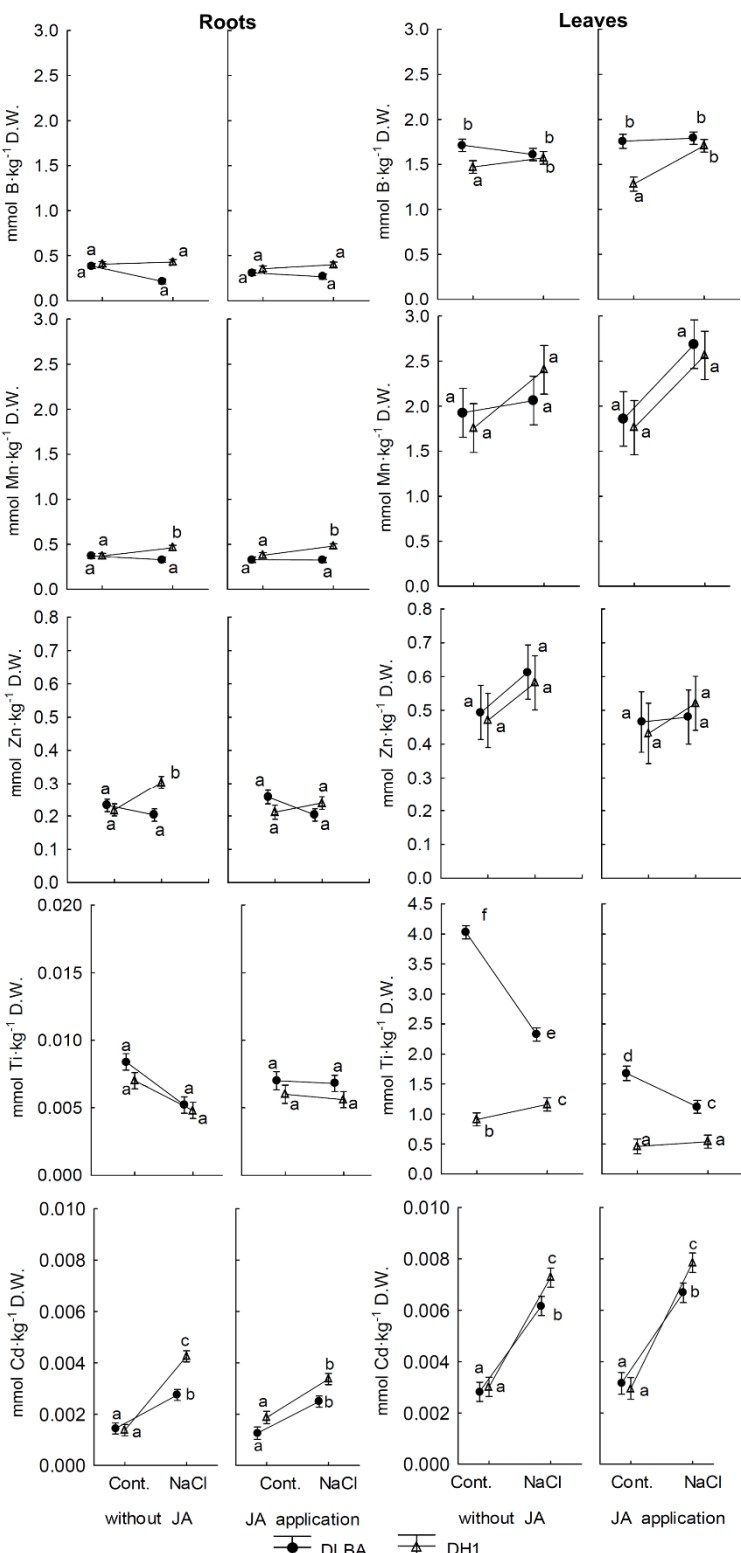

**Figure 3.** Concentrations of B, Mn, Zn, Ti and Cd in carrot storage roots and leaves. Means followed by the same letters are not significantly different at *p* = 0.05. Whiskers indicate standard error (*n* = 5).

In addition, a reduction in S content by 44% (Figure 2) and in Eu by 15% (Table S5) was observed in DLBA leaves to the levels found in the leaves of the control and stressed DH1 plants. In addition, the Ti content of DLBA leaves decreased by 34% under the influence of salt stress, but still remained at twice the level of DH1 leaves (Figure 3). The Ba content of the leaves decreased in both varieties by

44.0% in DLBA and 40.2% in DH1 (Table S3). In the roots of both varieties, the soil salinity caused a decrease in the Yb content, by 40% for DLBA and 50% for DH1 (Table S7).

There was no significant effect of salt stress on the concentrations of: (a) N, Ca, Mn, Cu, Mo, Al, Sr, Co, La, Li, Lu, Sm, Th, Y in the leaves or roots (Figures 2 and 3; Tables S3, S5 and S7); (b) Mg, Zn, Yb only in the leaves (Figures 2 and 3, Table S7); (c) S, B, Ti, Ba, Eu only in the roots (Figures 2 and 3, Tables S3 and S5).

The uptake of all the analyzed elements by single DLBA and DH1 plants /leaves + root/ was not significantly affected by the NaCl treatment (Table 3).

### 3.4. Interaction of Salinity and Foliar Application of JA on the Accumulation of other Elements in Plants

The foliar application of JA, considered independently of the other experimental factors, had little effect on the concentration of most of the elements in the leaves and roots of carrot plants (Tables 2 and 3, Figures 1–3 and Tables S3, S5 and S7). However, the concentrations of selected elements were modified by JA application depending on the variety and occurrence of salt stress ($p < 0.05$). In the roots of NaCl-treated DH1 plants, the foliar application of JA reduced the accumulation of Zn (by 21%) and Cd (by 20%) (Figure 3) and increased the accumulation of Yb by 68% (Table S7). By contrast, in both the NaCl-treated and control DLBA roots, JA had no effect on the concentration of these three elements (Figure 3). However, in the DLBA leaves, the application of JA caused a reduction in Ti content (by 58% and 46% in the control and NaCl-treated plants, respectively) (Figure 3) and a 60% increase in the Fe content in the leaves of control DLBA plants (Table S3).

In DH1 plants, after the application of JA, a significant reduction, by 44%, was found in the uptake of not only NaCl⁻, as stated above, but also of the sum of the concentrations of all the analyzed elements by a single plant (Table 3).

### 3.5. Concentrations of Elements in the Soil after Carrot Cultivation

After carrot cultivation, the soil was analyzed for the concentrations of the same elements that were determined in plants (Table 4 as well as Tables S1, S2, S4, S6 and S8). A significant influence of salinity and cultivated variety was found only in relation to the concentrations of Na, Cl and K in the soil after harvesting. The soil, after the cultivation of both carrot varieties treated with NaCl contained, on average, about 60 and 58 times more Na and 179 and 149 more Cl, for DLBA and DH1, respectively (Table 4). The concentration of K in the saline soil was significantly higher (in the range from 133% to 250%) than in the control, independently which variety was grown (Table S1).

**Table 4.** Concentrations (mmol·kg$^{-1}$) of Na and Cl in the soil after carrot harvest.

| JA Application | Variety | Treatment | Na | Cl |
|---|---|---|---|---|
| No JA | – | – | 29.98 ± 6.652a | 26.25 ± 6.007a |
| JA | – | – | 30.40 ± 6.545a | 30.45 ± 6.865a |
| – | DLBA | – | 31.87 ± 6.986b | 31.30 ± 7.077b |
| – | DH1 | – | 28.18 ± 6.191a | 25.19 ± 5.624a |
| – | – | Control | 0.95 ± 0.061a | 0.32 ± 0.011a |
| – | – | NaCl | 56.18 ± 1.644b | 53.37 ± 2.078b |
| No JA | DLBA | Control | 0.92 ± 0.154a | 0.33 ± 0.066a |
| No JA | DLBA | NaCl | 59.92 ± 2.848c | 54.87 ± 2.664c |
| No JA | DH1 | Control | 0.99 ± 0.176a | 0.30 ± 0.065a |
| No JA | DH1 | NaCl | 56.88 ± 2.572bc | 49.52 ± 2.664b |
| JA | DLBA | Control | 1.06 ± 0.099a | 0.33 ± 0.097a |
| JA | DLBA | NaCl | 59.41 ± 3.294c | 63.49 ± 2.664c |
| JA | DH1 | Control | 0.83 ± 0.051a | 0.34 ± 0.097a |
| JA | DH1 | NaCl | 48.53 ± 2.226b | 45.62 ± 2.664b |

Means followed by the same letters for main effects or 3-way interaction are not significantly different at $p = 0.05$; means ± standard error; $n$ = 5; JA—jasmonic acid.

### 3.6. Expression of the DcNHX4 Cation-Proton Exchange Transporter Gene

The *nhx4* gene expression was assessed based on PCR and qPCR using cDNA as the template. For DLBA leaves, clear PCR amplified fragments slightly longer than 300 bp were visualized in agarose gel that matched the expected 318 bp length of the product derived from the gene sequence while no PCR products were obtained for DH1 leaves (Figure 4). The expected fragment (95 bp) of the reference *actin-7* gene was amplified in both DLBA and DH1 leaves confirming good quality of cDNA from DH1. qPCR was then applied to verify whether *Dcnhx4* expression in DLBA is affected by salt stress. Expression analysis revealed that in salt-stressed DLBA plants, the *Dcnhx4* gene expression in leaves was 3.3-fold higher ($p = 0.006$) in comparison to plants growing in the control soil. In contrast, no expected PCR amplification products of *nhx4* were detected either in DLBA or DH1 roots. The expression of *Dcnhx4* in roots of the control and salt-stressed plants was also not confirmed by qPCR.

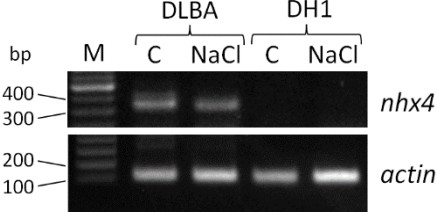

**Figure 4.** PCR-amplified products of the *nhx4* and *actin-7* genes in leaves of the control (C) and salt stressed (NaCl) DLBA and DH1 carrot plants; M—molecular weight marker.

## 4. Discussion

### 4.1. Response of Carrot Genotypes to Salinity Stress

Many species of cultivated plants respond to soil salinity by reduced biomass, including the biomass of the usable parts of crop plants [21,34]. Earlier reports have documented that salinity stress affected the length and diameter of carrot roots, its marketable and non-marketable yields of storage roots, and the mineral composition of the roots and leaves [35–37]. Our study confirmed that the stress of soil salinity (soil EC = 3.15 dS·m$^{-1}$) was a factor reducing plant growth and development, and consequently the biomass of carrot leaves and storage roots. Although both used varieties, DLBA and DH1, responded in biomass reduction their reaction was different (Figure 5). The DLBA variety is locally cultivated in the Fars region of Iran, where the soils are saline, which is a serious problem in agricultural production in that region [38]. DLBA exhibits morphologic features characteristic of the Eastern carrot type, i.e., it develops a tapering storage root and has pubescent leaves [20]. Both these morphologic features may help the plant to survive in conditions of water deficiency. A tapering root is usually long and can grow deeper into the soil, while pubescent leaves are adapted to exposure to intense sunlight. Leaves protected by hairs have a lower temperature and hence a lower transpiration rate [39]. The DH1 line was developed in a breeding program and originates from a carrot representing the Nantes type, the Western carrot type commonly grown in Europe and the USA. Such carrots are highly sensitive to salinity, which is manifested by the occurrence of leaf chlorosis, wilting, reduced growth and, finally, highly reduced biomass, in particular root biomass [21–23]. In our study, plants of both varieties were exposed to the same level of salt stress. The DLBA plants accumulated greater amounts of Na and Cl, with less biomass reduction, than the DH1 plants, that confirms the DLBA variety tolerates salinity better.

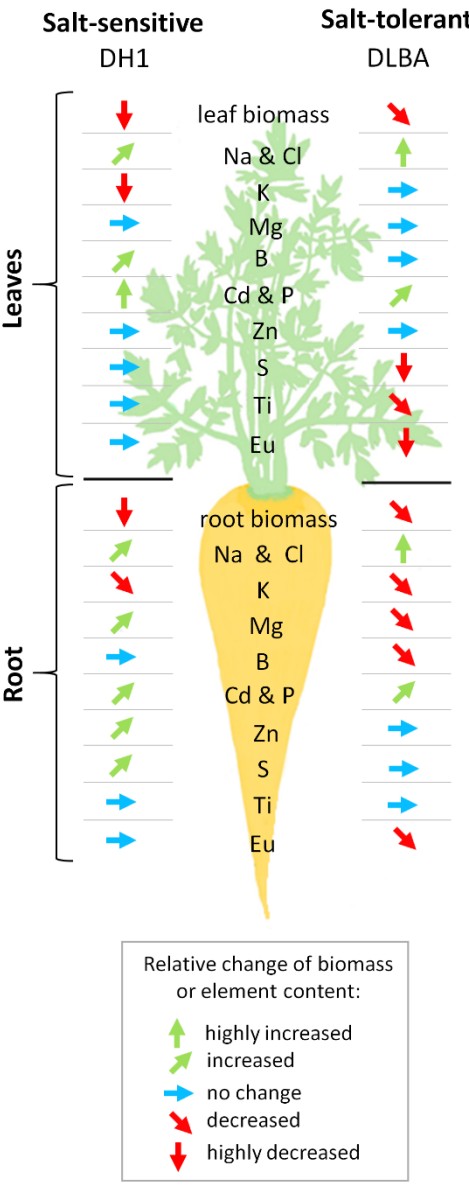

**Figure 5.** Schematic representation of changes in biomass and concentrations of elements in the leaves and storage root of salt-sensitive DH1 and salt-tolerant DLBA carrot plants. Arrows indicate an increase or decrease, or no change in element content in plants exposed to salt stress, in comparison with the control plants growing in non-saline soil. The angles of arrows indicate different intake levels for a given element.

We also observed that DLBA and DH1 had different strategies for mineral plant nutrition during salinity stress, which is presented in a schematic diagram in Figure 5. In general, the differences concerned: (**a**) the ability to Na and Cl uptake and their transport in the plant in the 'storage root-leaves' system; (**b**) the possibility of selective uptake of $K^+$ ions under conditions of high Na concentration; (**c**) the relationship between Cd and Zn in the 'soil-plant' system; (**d**) the uptake and transport of P, Mg, S, B and Ti in the 'roots-leaves' system.

### 4.2. Ability to $Na^+$ and $Cl^-$ Uptake and Their Transport in the Plant in the 'Storage Root-Leaves' System

The salinity stress increases the osmotic potential of the soil solution and limits water uptake by plants. In consequence, the passive transport of minerals along with the transpiration current is lowered, which may result in reduced uptake of minerals by roots with water. However, under the

conditions of salinity stress, carrot plants take up considerable amounts of Na and Cl elements, in addition they take up $Cl^-$ ions more intensely than $Na^+$ ions [23,35,37]. Such observations were also done in our study for both, salt-tolerant and salt-sensitive varieties, which indicates that the minerals were taken up also in a selective, active, manner by means of ion channels or protein carriers [2,40]. In both varieties, dry weight content in roots of salt-stressed plants slightly increased while decreased in leaves indicating that in such conditions carrot plants translocates more water to leaves. A higher accumulation of Na and Cl in leaves than in roots also indicates that the elements were transported from the roots to the leaves and the process mitigated element overload in the root. This process was more efficient in the salt-tolerant DLBA, which accumulated more Na and Cl in the leaves than the salt-sensitive DH1.

### 4.3. Possibility of Selective Uptake of $K^+$ Ions under Conditions of High Na Concentration

An antagonism between $K^+$ and $Na^+$ ions exists in plants growing under soil optimal conditions, and it increases greatly under salinity stress [2,41]. $Na^+$ ions strongly compete then with $K^+$ ions for uptake across the plant cell plasma membrane. This leads to a high $Na^+:K^+$ ratio in the tissues, which reduces plant growth due to $Na^+$ toxicity [42]. Golldack et al. [43] showed that under salt stress sensitive and tolerant rice lines differed in Na accumulation in leaves, but they both were able to maintain normal levels of K content. However, genes coding for $K^+$ transporters are differentially expressed in salt-tolerant and salt-sensitive plants [44] and the process is regulated in a tissue specific manner [45]. Transporters may be also differentially expressed in the roots, leaves and shoots during salt stress causing a strong reduction in $K^+$ uptake by the roots and, consequently, a decreased accumulation of this element in the leaves [46]. In a pot experiment, Ünlükara et al. [37] showed that watering carrots with saline water reduced the K content of the carrot plants. De Pascale & Barbieri [35] noted that the stress of NaCl salinity caused a reduction in the K content of carrot leaves to an extent greater than in the storage roots while simultaneously Na accumulation increased. In the field, with an increase in the NaCl dose (from 0% to 1% NaCl; w/v) there was a gradual decrease in the K content of the leaves.

The results of our study showed that under the influence of salt stress the K content of the roots of both carrot varieties decreased to a similar extent. In leaves, salinity decreased the K content in DH1, but did not in the salt-tolerant DLBA. These results indicate that DLBA has more efficient molecular mechanisms that enable active or selective uptake or transport of K from the roots to the leaves and at the same time it takes up larger amounts of Na and Cl.

The $Na^+:K^+$ ratio on the cell surface depends on the antagonistic relationships between $Na^+$ and $Ca^{2+}$, and increased $Na^+$ concentration may cause the removal of $Ca^{2+}$ from cell walls [47,48]. In rice, a reduced Ca level downregulated the expression of the gene encoding for an inward-rectifying $K^+$ channel that limited the uptake of $K^+$ [49]. In our study, however, a negative impact of NaCl treatment on the Ca content was not observed.

### 4.4. Expression of the DcNHX4 Cation-Proton Exchange Transporter Gene

In vitro studies have shown that in NaCl-stressed carrot cells, $Na^+$ cations were accumulated preferentially in the vacuole unlike $K^+$, which concentration in the cytosol doubled [50]. A high $Na^+$ concentration in cytosol is toxic thus cells transport $Na^+$ to vacuoles where its compartmentalization enables to maintain ion homeostasis in cytosol and osmotic balance [8]. NHX antiporters function as monovalent cation-proton exchangers and are responsible for a transmembrane $Na^+$ and $K^+$ transport. They are either localized to plasma membrane or to vacuole and endosomes, differ in their ion specificity, and play various roles in plant development, including response to ionic stress [10,51,52]. The expression of NHX genes is upregulated in saline conditions and NHX isoforms localized to vacuole are responsible for transport excess toxic ions out of cytosol to vacuolar lumen where they are accumulated [9,53]. The role of the class I cation-proton exchange transporter NHX4 isoform is however not fully revealed. Using a fluorescently tagged CFP-AtNHX4 fusion protein it was demonstrated that in *Arabidopsis* AtNHX4 is vacuole localized and its gene is induced by salt stress [54]. The same

authors indicated however that knockout of AtNHX4 led to a higher salt tolerance in *Arabidopsis* thus it was hypothesized that AtNHX4 transports $Na^+$ out of vacuole to cytosol [54]. These contradictory observations are explained by the fact that the role of NHX4 differs among species and highly depends on isoform variant [55].

In carrot, the *Dcnhx4* gene was annotated in gene prediction analysis of the DH1 genome within the reference carrot genome sequencing project [26]. We have identified that the *Dcnhx4* gene was expressed in DLBA leaves, but its transcripts were not detected in DH1, neither in the control nor salt-stressed plants. DH1 is a representative of Nantes type that has been bred for cultivation without selection pressure favoring genes, which expression may contribute to salt tolerance. The lack of *Dcnhx4* expression in DH1 suggests that NHX4 is not critical for carrot growth in common conditions. This conclusion is supported by reports demonstrating that a knockout of *Atnhx4* had no adverse effect on *Arabidopsis* biomass and growth when the mutants were growing in non-saline conditions [51,54]. Unlike DH1, DLBA variety grows in the region with saline soil where it is permanently exposed to such stress and selection pressure. Hence, the detected expression of *nhx4* in DLBA indicates that DcNHX4 may be essential for carrot plants to withstand in saline soil. Its role in carrot tolerance to salinity is further supported by the fact that *Dcnhx4* expression significantly increased when DLBA plants were exposed to salt stress. Furthermore, the *Dcnhx4* expression occurred in leaves, but not in roots that is in agreement with reports demonstrating organ-specific *Atnhx* expression [9,54].

NHX antiporters contribute in $K^+$ homeostasis in the cytosol of salt stressed cells by active transport of $Na^+$ and $K^+$ through tonoplast [51,56]. Comparison of *Arabidopsis* knockout mutants indicated that vacuolar uptake of both ions mediated by NHX4 depended on its splicing variants, which may significantly differ in their specificity to $Na^+$ and $K^+$ [55]. We observed that in leaves of DLBA exposed to salt stress, the K content remained unchanged in comparison to the control plants despite the Na content highly increased. In contrast, the K content decreased in DH1 leaves. These observations indicate that mechanisms of $Na^+$ transport in DLBA are more efficient which correlate with *Dcnhx4* expression. This may suggest that DcNHX4 has a higher specificity to $Na^+$ than to $K^+$. However, the hypothesis that DcNHX4 can also contribute to $K^+$ efflux from vacuolar lumen, as proposed by Li et al. [54] for AtNHX4 role in ensuring $K^+$ homeostasis in cytosol, cannot be excluded.

### 4.5. Relationship between Cd and Zn in the 'Soil-Plant' System

With increased concentration of chlorides in the soil, either as a result of fertilization with chloride fertilizers or due to soil salinity, chloro-complexation of cadmium (formation of cadmium ligands+chlorine) leads to the formation of Cd-chlorine speciation forms $[CdCl^{2-n}_n]$, which are easier taken up by plants than $Cd^{2+}$ cations are [57–59]. In addition, the chlorides alone may facilitate the diffusion of $Cd^{2+}$ into the cell apoplast in the roots [59,60]. The increased uptake and accumulation of Cd has been found in wheat [59,61], Swiss chard [59] and maize [62] when plants were exposed to salinity stress, although the limited uptake of Cd was also reported [63]. We suppose that, in our study, the increase in the accumulation of Cd in the roots and leaves of carrot plants under the influence of salinity stress was directly related to the effect of Cl on increasing the bioavailability of Cd to plants.

In wheat, salinity stress caused an increase in Zn uptake and its accumulation, and fertilization with Zn correlated with increased wheat tolerance to NaCl [64]. Zinc fertilization also caused a reduction in Cd accumulation in wheat [65]. Both elements can be transported with the same protein carriers, but plant species differ in their ability to uptake Zn and Cd [66–68]. Cd and Zn homeostasis is controlled by a complex molecular mechanism that is regulated, inter alia, by the reactive oxygen species (ROS) signaling pathway, and differences have been reported between metal-sensitive and metal-adapted plant species [67–69]. The results of our study indicate that there was a different functioning of plant mineral metabolism in the two carrot varieties in relation to the uptake and accumulation of these elements. In the DH1 carrot plants, the NaCl-induced stress caused increased accumulation of Cd in the leaves and roots, and a marked increase in Zn uptake and accumulation, mainly in the roots. In the DLBA plants changes in Cd accumulation were less evident and there was

no changes in Zn accumulation. Increased Cd uptake may be associated also with a lowering the nutritional status of plants with respect to Fe and Mn [70,71]. In our study, we did not find antagonistic relations between these elements and Cd and Zn.

### 4.6. Uptake by Plants and Transport of P, Mg, S, B, Ti in the 'Roots-Leaves' System

De Pascale and Barbieri [35] had found that the higher the NaCl dose used for plant watering (0–1% NaCl), the weaker the Fe transport from the carrot roots to the leaves. In addition, the increase in water salinity caused a gradual increase in the N and P contents of the leaves and storage roots; however, it did not affect the S, Ca and Mg contents of the leaves and storage roots. Inal et al. [72] showed that the concentration of P in the shoots and roots of carrot and of S, Mg and Si in the roots were not significantly affected by salinity treatments, while the NaCl salinity reduced S and Si, and increased Mg concentrations in the shoots. Furthermore, Fe, Zn, Mn and Mo concentrations in the shoots did not change under the salinity stress. Kleiber [73] found that the increasing concentration of Ti (under Mn stress) in lettuce plants was strongly negatively correlated with the Fe content and to a lesser extent negatively correlated with decreasing P, Mg and Cu contents. The effect of NaCl on the concentrations of N, P, K, Ca, Cu, Zn, Mn and Fe has been reported in other species, too [74,75].

In our study, we too noted an antagonistic relationship between the P and Ti contents, but only in DLBA. Under the NaCl-induced stress, there was a decrease in the Ti content of the leaves, which was associated with an increase in the P content of the roots and leaves. In salt-sensitive DH1, the Ti *versus* P relations were different, the Ti content of the leaves and roots did not change considerably while the P content increased significantly.

Phosphorus is involved in many physiological and biochemical processes of plants. It is, among other, an energy carrier in photosynthesis and respiration processes. Magnesium, in turn, is a cofactor for many enzymes and is present in the porphyrin system of chlorophyll molecules [40]. Ünlükara et al. [37] showed that watering with saline water reduced the concentrations of Mg and also Ca in carrot plants at different salinity thresholds, 1.13 and 5.0 dS·m$^{-1}$, respectively. In our study, in the roots of DH1 under the influence of NaCl stress, the uptake of P and Mg was more intense than in DLBA. In the leaves, in turn, increased accumulation of P was observed. No changes to Ca accumulation were observed in either carrot variety used, but the plants were growing in the soil with EC lower than the above mentioned threshold. These observations indirectly indicate that in DH1 there must have occurred an intensification in the biochemical and physiological processes of defense/adaptation to stress conditions that required increased P and Mg uptake. They may have been associated with the intensification of photosynthesis, which is commonly observed under conditions of mild salinity stress [76].

### 4.7. Influence of JA on the Mineral Balance in Plants

Positive effects of exogenous JA on the growth or physiological and biochemical processes of plants subjected to salt stress have been found, for example, in wheat [15], rice [77], pea and barley [78] and basil [79]. Velitchkova and Fedina [80] had shown that JA had an effect on reducing Na+ and Cl$^{-}$ uptake by pea plants.

In our study, the foliar application of exogenous JA had a significant effect only on the functioning of some parameters of the carrot plant mineral metabolism, but did not affect the amount of biomass of DH1 and DLBA plants. However, in the plants subjected to salt stress, the foliar application of JA improved the functioning of plant mineral metabolism in the sensitive DH1 to a greater extent than it did in the tolerant DLBA. As a result, the DH1 plants had reduced uptake of Na and Cl by the storage root, leaves and the whole plant. Similar reaction of salt-sensitive rice plants was reported earlier [81]. The sum of the uptake of all the analyzed elements was also reduced in the DH1 carrot plants. In the roots of this variety, Zn and Cd accumulation was reduced while Yb accumulation increased. We can speculate that the tolerance of DLBA plants may have been associated with a higher level of endogenous JA, which was involved in the response to salt stress, and whose level in the

sensitive DH1 plants was insufficient. This kind of correlation had been observed in tomato where varieties tolerant to salt stress had higher levels of jasmonates compared with sensitive varieties [82]. Annuality of DLBA observed in our study further supports the hypothesis on higher JA activity in this variety as jasmonates participate in the regulation of flowering processes [83]. Whether there really are differences in the concentrations of JA or derivatives of this phytohormone between sensitive and tolerant carrot plants requires further research.

## 5. Conclusions

The salinity stress affected the growth and mineral nutrition of two carrot varieties, salt-sensitive DH1 and salt-tolerant DLBA. The DLBA plants accumulated larger amounts of Na and Cl in the roots and had an increased transport of these elements to the leaves than DH1 plants, but an adverse effect of salinity stress on DLBA plant growth was less evident than in DH1. This indicates that DLBA has a more effective mechanism of detoxification of high Na and Cl concentrations in plant tissues. The expression of DcNHX4 cation:proton exchange transporter indicates more efficient monovalent cation compartmentalization in the DLBA leaves that corresponds to a stable uptake of K by DLBA despite the presence of high Na concentration in leaves. Both varieties differed also in their ability to uptake and accumulate other elements, such as Mg, Zn, S, Cd, P and B, in an organ-specific manner. Exogenous JA had very little effect on DLBA plant growth and the functioning of mineral metabolism, but it contributed to reducing the uptake of Na and Cl as well as the sum of the uptake of all the analyzed elements by DH1 plants. Also changes in the accumulation of some other elements were observed in the roots of DH1 plants. This indicates that JA have an impact on mechanisms that may be involved in tolerance to salinity, but only in the salt-sensitive carrot variety.

**Supplementary Materials:** The following are available online at http://www.mdpi.com/2073-4395/10/5/659/s1, Table S1: Concentrations of N, P, K, Mg, Ca and S in the soil after carrot cultivation (mmol·kg$^{-1}$); Table S2: Concentrations of B, Mn, Zn, Cd and Ti in the soil after carrot cultivation (mmol·kg$^{-1}$); Table S3: Concentrations of Cu, Fe, Mo, Al and Ba in carrot leaves and storage roots (mmol·kg$^{-1}$ D.W.); Table S4: Concentrations of Cu, Fe, Mo, Al and Ba in the soil after carrot cultivation (mmol·kg$^{-1}$); Table S5: Concentrations of Sr, Co, Eu, La and Li in carrot leaves and storage roots (mmol·kg$^{-1}$ D.W.); Table S6: Concentrations of Sr, Co, Eu, La and Li in the soil after carrot cultivation (mmol·kg$^{-1}$); Table S7: Concentrations of Lu, Sm, Th, Y and Yb in carrot leaves and storage roots (mmol·kg$^{-1}$ D.W.); Table S8: Concentrations of Lu, Sm, Th, Y and Yb in the soil after carrot cultivation (mmol·kg$^{-1}$).

**Author Contributions:** S.S., M.K.-C. and R.B. conceived and designed experiments; M.K.-C. and A.L. performed experiments and gene expression analyses; S.S., performed element content analyses, data analysis and wrote the manuscript; R.B. and A.L. revised the manuscript. All authors have read and agreed to the published version of the manuscript.

**Funding:** This research was financed by the National Science Centre, Poland (grant UMO-2016/21/B/NZ9/01054).

**Acknowledgments:** The authors thank Alicja Macko-Podgorni for her advice on gene expression analysis.

**Conflicts of Interest:** The authors declare no conflict of interest.

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
