# Peer review of "Effect of Soil Salinity and Foliar Application of Jasmonic Acid on Mineral Balance of Carrot Plants Tolerant and Sensitive to Salt Stress"

_agronomy, doi:10.3390/agronomy10050659_

Round 1

Reviewer 1 Report

I have read the manuscript entitled "Mineral balance in carrot plants tolerant and sensitive to soil salinity". It is well presented and provides useful information about the effects of soil salinity stress and foliar application of jasmonic acid on the mineral balance in a salt-sensitive and a salt-tolerant carrot plants. However, there are some concerns with respect to the manuscript should be addressed before it can be published.

MATERIAL AND METHODS.

  • Why have you chosen JA application and no other hormones such as SA? Previous studies proved that the application of salicylic acid (SA) improved the growth of plants under salt stress.

RESULTS.

  • In my opinion, Table 1 and 3 could be more understandable in a bar graph representation.

DISCUSSION

  • This part needs to be streamlined.
  • In line 565, you mentioned “we can speculate that the tolerance of DLBA plants might have been associated with a higher level of endogenous JA”. Could you measure it in order to confirm this hypothesis?

Author Response

Author's reply to the Review Report (Reviewer #1)

Dear Editor and Reviewer #1

We would like to thank the Editor and Reviewer for their time and efforts put into thorough reading and the review on the manuscript as well as giving us comments and suggestions how to improve our manuscript. We have found all comments valuable and have implemented the corrections according to suggestions. We hope that all the changes and improvements we have introduced into the revised manuscript will meet journal requirements and Editor acceptance. Modified parts in the revised manuscript are marked in blue.

Please, find below a point-by-point response to the Reviewer’s comments

In the review:

„Comments and Suggestions for Authors
”

I have read the manuscript entitled "Mineral balance in carrot plants tolerant and sensitive to soil salinity". It is well presented and provides useful information about the effects of soil salinity stress and foliar application of jasmonic acid on the mineral balance in a salt-sensitive and a salt-tolerant carrot plants. However, there are some concerns with respect to the manuscript should be addressed before it can be published.

MATERIAL AND METHODS

Why have you chosen JA application and no other hormones such as SA? Previous studies proved that the application of salicylic acid (SA) improved the growth of plants under salt stress.”

Our answer:

The role of SA in the induction of plant tolerance to biotic and abiotic stress is indeed described in the literature. The effect of JA is much less known, however, its beneficial effect on the development of plants exposed to salinity stress has been demonstrated so far, e.g. in rice, strawberry, soybean, and we included a relevant information in the Introduction section (L 58-71). The role of JA has not be evaluated in carrot thus we sought to include novel aspect in our research. This decision was stimulated by our preliminary experiments carried out using in vitro cultures. We found that JA might induce tolerance of carrot seedlings when seeds were exposed to saline mineral medium supplemented with JA. For these reasons we decided to verify JA effect in plants growing in saline soil and planned a relevant experiment in a grant proposal as described in this paper.

In the review:

RESULTS

In my opinion, Table 1 and 3 could be more understandable in a bar graph representation.”

Our answer:

Thank you for this comment which we thoroughly considered. Reviewer #2 had also comments about the way the results were presented in these tables but they recommended to add additional results concerning the main effects. More results can be presented when arranged in a table with modified layout, but it seems not possible to make a graphical presentation of all these results in a readable way. Moreover, the tables contain variables with values that differ in magnitude of several times and separate scales would have to be applied to visualize them to reveal differences among treatment combinations. With three factors to be compared, this approach seems an option that would not guarantee better understanding and may cause confusion to readers. Taking into account suggestions of both Reviewers, we incline to extend information provided in tables. We do hope the changes we provided in the revised tables and our explanation is satisfactory. 

In the review:

DISCUSSION

This part needs to be streamlined.

Our answer:

Thank you for this comment. We have revised the discussion according to your suggestion and have improved it. The text has been shorten by about 25% by removing some redundant statements and descriptions of relatively low importance. Large fragments of the text has been rewritten, including the whole Conclusions section as requested also by Reviewer #2. In our opinion the revised text is now more relevant, fluent and concise thus we hope that such improved Discussion meets expectations.

In the review:

In line 565, you mentioned “we can speculate that the tolerance of DLBA plants might have been associated with a higher level of endogenous JA”. Could you measure it in order to confirm this hypothesis?”

Our answer:

Indeed, the confirmation of differences in JA content between two varieties used would be appreciated. This is in particular a tempting research task as analogous differences were found in tomato what is mentioned in our discussion (L 552-559 in the revised text). We wish to include such analyses in future experiments if we can.

We hope that all the changes and improvements are satisfied. We also declare our willingness to make  further improvements if such are identified by the Editor or Reviewers.

We declare that the manuscript has not been submitted for publication elsewhere.
All co-authors have contributed to this article and all agree to submit it into the Agronomy journal (ISSN 2073-4395). There are no conflicts of interests.

We would be grateful for the acceptance of our manuscript for publication in Agronomy.

Yours sincerely,

On behalf of the Authors,

Dr.Sc. Sylwester Smoleń, Associate professor

Unit of Plant Nutrition

Department of Plant Biology and Biotechnology
Faculty of Biotechnology and Horticulture, University of Agriculture in Krakow

Al 29 Listopada 54

31-425 Kraków

POLAND

sylwester.smolen@urk.edu.pl   Sylwester.Smolen@interia.pl

Reviewer 2 Report

The manuscript “Mineral balance in carrot plants tolerant and sensitive to soil salinity” presents the effects of soil salinity stress and foliar application of jasmonic acid (JA) on the mineral balance in salt-sensitive (DH1) and salt-tolerant 10 (DLBA) carrot varieties. The salt tolerance mechanisms and the strategies to increase vegetable tolerance to salinity are interesting topics.

Nevertheless, the manuscript has one main issue: the experiment tested the effects of 3 factors (JA, salt stress and carrot varieties) on plant biomass and mineral uptake and translocation but the results do not present separately the main effects of the factors and all the interactions among the factors.

Moreover, the material and methods section lacks many useful details.

These issues should be solved before the manuscript could be considered suitable for publication.

Specific comments

Title

I suggest to modify the title by adding the use of jasmonic acid in order to be more self-explanatory and recall the aim of the paper, i.e. “Effect of jasmonic acid and soil salinity on mineral balance of carrot plants”

Abstract

L.10-11 “…in salt-sensitive (DH1) and salt-tolerant 10 (DLBA) carrot plants.”

Add some information on the effect of JA on root and leaf biomass.

Introduction

L.66-67 define MDA, SOD, POD and GSH

Material and methods

Add biomass sampling and determination

L.103 Where was conducted this experiment (location, coordinates, altitude)? Which were the climatic conditions of the site and inside the tunnel?

L.103-104 96 L; Add all the dimensions of the containers

L.104 Characteristics and commercial name and producer of peat substrate?

L.105 Specify the size and characteristics of the tunnel(s?)

L.107 Which were the characteristics of the soil used to cover the substrate?

L.109 “regularly watered” is too undefined, please better specify the watering interval; Add the EC of saline solution and tap water used for irrigation

L.110 0.6 L

L.112-113 It’s not clear if the containers of control plants were filled with the substrate and covered with a soil layer (as saline trial) or if they were filled only with soil as stated

L.117 Specify the supplier of JA

L.122 add the amount of the nutrient elements in Yaraliva as done for nutrifol

L.128-129 Which was the statistical design adopted? It’s not clear if each replication corresponded to one container (13 plants / replication ?) and how plants were distributed among treatments.

L.131-141 Define the sample size (n. of plants, weight of dry sample, n. of replicates for each analysis)

L.142-143 As you used both substrate (peat + sand) and soil in the experiment, you should better describe what you have sampled for chemical analysis

L.144 Describe soil samples extraction or add a reference

L.147, 149  by the ICP-OES technique… add references

L.147 The extraction was performed on soil samples or on extracts obtained with acetic acid? Not clear.

L.159 Bio-Rad… add city and Country

L.161 A&A Biotechnology add city and Country

L.168 …add city and Country

L.170 add city and Country

L.174 add city and Country

Results

In this experiment you tested 3 factors (salt stress, variety, JA) but in the tables you only report the data of the interaction NaClxVarietyxJA. You should also show the effect of the main factors and the simple interactions and at least if the simple interactions (NaClxVariety, NaClxJA, VarietyxJA) was significant.

L.193-208 Authors only show the effect of the interaction NaClxVarietyxJA on plant fresh biomass. NaCl may affect water uptake and translocation, thus it would be interesting to show also the effect of the experimental factors on plant dry biomass. Please add this parameter to the results.

Discussion

L.388,510  De Pascale

Conclusion

Try to resume the main findings of the experiment in a more concise way.

Author Response

Author's Reply to the Review Report (Reviewer #2)

Dear Editor and Reviewer #2

We would like to thank the Editor and Reviewer for their time and efforts put into thorough reading and the review on the manuscript as well as giving us comments and suggestions how to improve our manuscript. We have found all comments valuable and have implemented the corrections according to suggestions. We hope that all the changes and improvements we have introduced into the revised manuscript will meet journal requirements and acceptance. Modified parts in the revised manuscript are marked in blue.

Please, find below a point-by-point response to the Reviewer’s comments

In the review:

The manuscript “Mineral balance in carrot plants tolerant and sensitive to soil salinity” presents the effects of soil salinity stress and foliar application of jasmonic acid (JA) on the mineral balance in salt-sensitive (DH1) and salt-tolerant 10 (DLBA) carrot varieties. The salt tolerance mechanisms and the strategies to increase vegetable tolerance to salinity are interesting topics.

Nevertheless, the manuscript has one main issue: the experiment tested the effects of 3 factors (JA, salt stress and carrot varieties) on plant biomass and mineral uptake and translocation but the results do not present separately the main effects of the factors and all the interactions among the factors.

Moreover, the material and methods section lacks many useful details.

These issues should be solved before the manuscript could be considered suitable for publication.

 Our answer:

We would like to thank the Reviewer for these comments that we have found valuable and we appreciate the Reviewer gave us detailed suggestions how to improve the manuscript. As the above raised points are also listed by the Reviewer in specific comments we answer to them below, point by point in the order they appear in the review.

Specific Reviewer’s comments are in italics

Title

I suggest to modify the title by adding the use of jasmonic acid in order to be more self-explanatory and recall the aim of the paper, i.e. “Effect of jasmonic acid and soil salinity on mineral balance of carrot plants”

Our answer:  We thank the Reviewer for this suggestion. We agree that JA effect on carrot response to salinity stress should be included in the title as it is one of the objective of our research. Accordingly, we modified the title so the revised version includes both salinity stress and JA application although in reversed order than suggested.

Abstract

L.10-11 “…in salt-sensitive (DH1) and salt-tolerant 10 (DLBA) carrot plants.”

Our answer:  Corrected

Add some information on the effect of JA on root and leaf biomass

Our answer:  Additional information has been included in the Abstract (L. 21-23).

Introduction

L.66-67 define MDA, SOD, POD and GSH

Our answer:  Compound full names are provided now (L. 68-69).

Material and methods

Add biomass sampling and determination

Our answer:  We added additional description how plants were collected and their mass determined (L. 139-143).

L.103 Where was conducted this experiment (location, coordinates, altitude)? Which were the climatic conditions of the site and inside the tunnel?

Our answer:  Experiment location with coordinates and altitude is provided now (L. 107-110). We have not recorded climatic conditions throughout vegetation period, however the plastic tunnel had an tilted side vent in the roof that automatically opened when temperature inside the tunnel reached 18 oC/12 oC (day / night) to prevent overheating the plants.

L.103-104 96 L; Add all the dimensions of the containers

Our answer:  Corrected; the dimensions are provided now (L. 110).

L.104 Characteristics and commercial name and producer of peat substrate?

Our answer:  The producer is provided now (L. 112).

L.105 Specify the size and characteristics of the tunnel(s?)

Our answer:  The size of the tunnel and the presence of tilted side vent in the roof is added (L. 106-107).

L.107 Which were the characteristics of the soil used to cover the substrate?

Our answer:  The properties of soil including, pH, EC, Eh and macroelement contents are listed now (L. 112-114).

L.109 “regularly watered” is too undefined, please better specify the watering interval; Add the EC of saline solution and tap water used for irrigation

Our answer:  Watering was done every 2-3 days and such information has been included in the revised text (L. 119). We have also provided EC of tap water (0.6 dS·m-1) and NaCl solution (10.7 dS·cm-1) (L. 118-120).

L.110 0.6 L

Our answer:  Corrected

L.112-113 It’s not clear if the containers of control plants were filled with the substrate and covered with a soil layer (as saline trial) or if they were filled only with soil as stated

Our answer:  Only one peat substrate:sand mixture was used in our experiment. We realized that we used two terms to name this mixture that was certainly misleading. In the revised manuscript we have defined the term soil as a mixture of peat and sand (L. 111-116) and have used this term consistently throughout the revised manuscript. Hence, containers were filled up to 25 cm in height with the soil of EC either 0.2 dS·m-1 or 3.0 dS·m-1 (soil mixed with NaCl) and then such soil was covered with extra 10 cm layer of the same soil with EC 0.2 dS·cm-1.

L.117 Specify the supplier of JA

Our answer:  Supplier is provided now (L. 126).

L.122 add the amount of the nutrient elements in Yaraliva as done for nutrifol

Our answer:  Nutrient composition is provided now (L. 132).

L.128-129 Which was the statistical design adopted? It’s not clear if each replication corresponded to one container (13 plants / replication ?) and how plants were distributed among treatments.

Our answer:  All treatment combinations were randomly arranged and the experiment was conducted in five replications. One replication corresponded to one container with 13 plants. We have modified the text to be more clear (L. 135-137).

L.131-141 Define the sample size (n. of plants, weight of dry sample, n. of replicates for each analysis)

Our answer:  Such details are provided now (L. 145-146, 156-157).

L.142-143 As you used both substrate (peat + sand) and soil in the experiment, you should better describe what you have sampled for chemical analysis

Our answer:  As described above, only one kind of soil was used. After text modification and consistent use of the term “soil” we assume that the description is clear now (L. 159).

L.144 Describe soil samples extraction or add a reference

­Our answer:  Soil extraction is described now (L. 159-161).

L.147, 149  by the ICP-OES technique… add references

­Our answer:  The citation to reference is added (L. 151).

L.147 The extraction was performed on soil samples or on extracts obtained with acetic acid? Not clear.

Our answer:  Measurements were performed on extracts. The description of the procedure has been modified accordingly (L. 159-161).

L.159 Bio-Rad… add city and Country

L.161 A&A Biotechnology add city and Country

L.168 …add city and Country

L.170 add city and Country

L.174 add city and Country.”

Our answer:  All details are provided now (L. 175, 177, 184, 187, and 191).

Results

In this experiment you tested 3 factors (salt stress, variety, JA) but in the tables you only report the data of the interaction NaClxVarietyxJA. You should also show the effect of the main factors and the simple interactions and at least if the simple interactions (NaClxVariety, NaClxJA, VarietyxJA) was significant.

Our answer: We thank the Reviewer for these comments that we has found very valuable. In the original manuscript we presented results as 3-way breakdown tables to show all effects that might outcome from interaction of all factors. In that way we showed all partial means and independent on the tables we also described main effects and their interactions in the text. After considering the Reviewer’s comment, we agree that the presentation of main effects was not enough exposed and the construction of sentences might cause difficulties to identify which interactions were significant. Therefore, according to the Reviewer’s suggestion we have extended tables 1-4 and included main effects there. We have also modified the text in the Results sections and included description of main effects and their interactions followed by the results of ANOVA indicating whether they were statistically significant or not. These changes are implemented in sections 3.1-3.5.

L.193-208 Authors only show the effect of the interaction NaClxVarietyxJA on plant fresh biomass. NaCl may affect water uptake and translocation, thus it would be interesting to show also the effect of the experimental factors on plant dry biomass. Please add this parameter to the results.

Our answer:  Dry weight is provided in Table 1 now. These results are explained in the text and the differences observed additionally commented in the Discussion section (L. 214-216, 226-229, 400-402).

Discussion

L.388,510  De Pascale

Our answer:  corrected (L. 426, 684)

Conclusion

Try to resume the main findings of the experiment in a more concise way.

Our answer:  We appreciate this remark very much. We have rewritten the Conclusions section and restricted it by about 50% to show the most important findings (L. 561-574).

 Additionally, we would like to acknowledge that according to the Reviewer’s #1 advice we have reduced the Discussion section by about 25% to make it more relevant, fluent and concise.

We hope that all the changes and improvements are satisfied. We also declare our willingness to make  further improvements if such are identified by the Editor or Reviewers.

We declare that the manuscript has not been submitted for publication elsewhere.
All co-authors have contributed to this article and all agree to submit it into the Agronomy journal (ISSN 2073-4395). There are no conflicts of interests.

We would be grateful for the acceptance of our manuscript for publication in Agronomy.

Yours sincerely,

On behalf of the Authors Team,

Dr.Sc. Sylwester Smoleń, Associate professor

Unit of Plant Nutrition

Department of Plant Biology and Biotechnology
Faculty of Biotechnology and Horticulture, University of Agriculture in Krakow

Al 29 Listopada 54

31-425 Kraków

POLAND

sylwester.smolen@urk.edu.pl   Sylwester.Smolen@interia.pl

Round 2

Reviewer 1 Report

The changes and improvements have improved the manuscript. 

Reviewer 2 Report

The manuscript has been significantly improved according to reviewer's suggestions but still needs a deep editing of English language and style